# Hematological abnormalities and comorbidities are associated with COVID-19 severity among hospitalized patients: Experience from Bangladesh

**Md. Ashrafur Rahman**[1], **Yeasna Shanjana**[2], **Md. Ismail Tushar**[1], **Tarif Mahmud**[1], **Ghazi Muhammad Sayedur Rahman**[1], **Zahid Hossain Milan**[1], **Tamanna Sultana**[1], **Ali Mohammed Lutful Hoq Chowdhury**[3], **Mohiuddin Ahmed Bhuiyan**[4], **Md. Rabiul Islam**[4]*, **Hasan Mahmud Reza**[1]*

1 Department of Pharmaceutical Sciences, North South University, Bashundhara, Dhaka, Bangladesh,
2 Department of Environmental Sciences, North South University, Bashundhara, Dhaka, Bangladesh,
3 Critical Care Department, Evercare Hospital, Bashundhara, Dhaka, Bangladesh, 4 Department of Pharmacy, University of Asia Pacific, Farmgate, Dhaka, Bangladesh

* robi.ayaan@gmail.com (RI); hasan.reza@northsouth.edu (HMR)

**Data Availability Statement:** All relevant data are within the manuscript.

## Abstract

### Background

The hematological abnormalities are assumed to be involved in the disease progression of COVID-19. However, the actual associations between specific blood parameters and COVID-19 are not well understood. Here we aimed to assess the correlations between hematological parameters and the severity of COVID-19.

### Methods

We included COVID-19 patients who were admitted to Evercare Hospital Ltd, Dhaka, Bangladesh, between November 10, 2020, to April 12, 2021, with a confirmed case of RT-PCR test. We recorded demographic information, clinical data, and routine hematological examination results of all COVID-19 patients. We performed statistical analyses and interpretation of data to compare severe COVID-19 patients (SCP) and non-severe COVID-19 patients (NSCP).

### Results

The age and BMI of the admitted COVID-19 patients were 48.79±8.53 years and 25.82 ±3.75 kg/m$^2$. This study included a total of 306 hospitalized COVID-19 patients. Among them, NSCP and SCP were 198 and 108, respectively. And we recorded 12 deaths from SCP. We observed the alterations of several hematological parameters between SCP and NSCP. Among them, we noticed the increased levels of C-reactive protein (CRP), d-dimer, and ferritin showed good indicative value to evaluate the severity of COVID-19. Also, there were positive correlations among these parameters. Moreover, we found correlations between the outcomes of COVID-19 patients with patient's demographics and comorbid diseases.

**Funding:** The author(s) received no specific funding for this work.

**Competing interests:** The authors have declared that no competing interests exist.

## Conclusion

Based on our results, CRP, d-dimer, and ferritin levels at admission to hospitals represent simple assessment factors for COVID-19 severity and the treatment decisions at the hospital setup. These blood parameters could serve as indicators for the prognosis and severity of COVID-19. Therefore, our study findings might help to develop a treatment protocol for COVID-19 patients at the hospital setup.

## Introduction

The outbreak of the COVID-19 due to the SARS-CoV-2 was firstly epi-centered in Hubei province, Wuhan, China. The SARS-CoV-2 is a high transmissible virus that spread across the world within a short period [1, 2]. Therefore, on March 12, 2020, the world health organization (WHO) declared COVID-19 a pandemic for the world. Globally, the ongoing pandemic has created numerous challenges to the healthcare systems of many countries [3]. The risk of coronavirus infection and the development of severe COVID-19 is associated with age and comorbid diseases of patients [4]. The COVID-19 primarily targets the lungs of the patients. However, it can damage other organs in severe cases. Therefore, hematological abnormalities and their intensities are involved in the severity of COVID-19. About half of the COVID-19 patients are asymptomatic carriers and pre-symptomatic. The fever, cough, sore throat, fatigue, dyspnea, myalgia, and breathing problems are the initial symptoms of COVID-19. These symptoms may progress to acute respiratory distress syndrome (ARDS), metabolic acidosis multi-organ failure, and shock in severe COVID-19 cases. These worsen conditions of patients may lead to death [5, 6].

COVID-19 plays a critical role in the host immune response in disease pathogenesis and clinical manifestation. It triggers the antiviral immune system to produce an uncontrolled inflammatory response that may lead to cause hematological abnormalities such as lymphopenia, lymphocytes, and abnormalities in granulocyte and monocyte [7]. These hematological abnormalities can facilitate the infection by other microorganisms, septic shock, and multi-organ dysfunction. Moreover, individuals with comorbid diseases such as hypertension, obesity, diabetes, etc., are at higher risk of coronavirus infection. Most comorbidities are frequently associated with excessive body fat mass [8]. As a result, it alters various hormonal, metabolic, and inflammatory functions. Also, adipose tissue may involve in the pathogenesis of COVID-19, where obesity plays an important role [9]. Hypertension is also a risk factor for coronavirus infection. COVID-19 patients with hypertension might increase severity and mortality rates. Hypertension should consider as a clinical predictor for the severity of COVID-19 both in adults and children [10]. Also, diabetes is another comorbidity to increase the COVID-19 disease burden and mortality worldwide [11, 12]. It plays a primary role in the development of pneumonia and sepsis due to the virus infection. A study showed that diabetic patients are at higher risk of coronavirus infection. Also, they have an increased chance of COVID-19 severity and mortality [13]. However, some COVID-19 patients with comorbid diseases are recovering. Unfortunately, it might create an increased burden for them due to post-COVID complications.

We have seen increasing evidence about hematological abnormalities and comorbidities with the severity of COVID-19 and duration of hospitalization. Still, the elaborative studies regarding the relationship of hematological abnormalities with the severity and extended hospitalization of COVID-19 patients are limited in different countries. Also, the severity and mortality of coronavirus infection vary from country to country due to the local mutation of the virus in various geographical locations [14]. Thus, most studies have limitations according

to geographical consideration, parameters analyzed, and prognostic value of altered elements. Moreover, Bangladesh is a densely populated country with a significant number of aged populations [15, 16]. In Bangladesh, 7.7% of the total population is over 60 years, and 53.8% of elderly adults have comorbidities such as blood disorders, hypertension, diabetes, COPD, etc. [17–21]. Also, the inadequate number of health care service providers and limited healthcare facilities making the situation more destructive [22]. Therefore, we aimed to assess the associations of hematological parameters and comorbid diseases with the severity levels of COVID-19 patients in Bangladesh.

## Materials and methods

### Study population

This observational cohort study screened 306 COVID-19 patients for hematological abnormalities and comorbid diseases from the COVID unit of Evercare Hospital, Dhaka, Bangladesh, from November 10, 2020, to April 12, 2021. The present study included hospitalized COVID-19 patients with multiple complications. Inclusion criteria were COVID-19 patients over 18 years of age with a confirmed PCR diagnosis. The severity of COVID-19 patients was assessed by: 1. patients admitted in hospital with confirmed pneumonia by lung CT, 2. patients with respiratory distress (rate > 30 breaths/min) and oxygen capacity level < 93%, and 3. patients required intensive care unit (ICU) support or mechanical ventilation [23, 24]. Exclusion criteria of this study were less than 18 years old, pregnant women and those who are taking medicine for reducing lymphocyte, leukocytes, or white blood cells count, and patients previously diagnosed with any hematological disorders.

### Sample collection and analysis

Ten milliliters of blood samples were collected using an ethylenediaminetetraacetic acid-containing vacutainer tube for analysis of hematological parameters. We monitored the diet restrictions for 6 hours before taking the samples from patients. The puncture site was also adequately sanitized and cleaned before sample collection. Immediately after blood collection, the routine procedures were completed and the samples were stored at -80˚C [25–28].

Several laboratory tests were performed during routine examination of admitted COVID-19 patients at the hospital. We collected data for the hematological parameters of COVID-19 patients at different severity stages. The immunoassay was performed using Beckman Coulter Access-2 and Sysmex CS-1600 to measure ferritin and d-dimer levels, respectively. Also, the same immunoturbidimetry measured the C-reactive protein (CRP) levels using Beckman Coulter, DXE 700AU. The photometric method was applied using Beckman Coulter, DXE 700AU for creatinine, liver function tests, Pro-BNP, and Sysmex XN 2000 for hemoglobin (Hb) and complete blood count (CBC). White blood cell (WBC) concentration was detected by flow cytometry method using Sysmex XN 2000. The flow detection method applied using Sysmex XN 2000 for red blood cell (RBC) and platelet count test. The ion-selective electrode method analyzed electrolyte parameters ($Na^+$, $Cl^-$, $K^+$, and $HCO_3$) using Dimension EXL 200. The chemicals and reagents of analytical grade were collected from commercially available companies to perform the above tests. Also, all the standard samples were purchased from Sigma-Aldrich, Inc.

### Statistical analysis

We presented the hematological parameters as the mean ± standard deviation (mean ± SD). Here we compared the changes of parameters between severe COVID-19 patients (SCP) and

non-severe COVID-19 patients (NSCP) using independent sample t-tests. We assessed the correlations among different blood parameters using Pearson's correlation test. We measured the risk factors of COVID-19 severity using univariate and multivariate analyses. We evaluated the diagnostic performance of the studied parameters using receiver operating characteristic (ROC) analysis. We performed data editing, sorting, coding, classification, and tabulation using Microsoft Excel. also, we applied IBM SPSS software (version 25.0) for all statistical analyses. We considered significant statistical variations or associations at a p-value less than 0.05.

## Ethics

The Institutional Review Board (IRB) of North South University (IRB number: 2020/OR-NSU/ IRB-No.0701) approved the study protocol. We performed the investigations according to the principles stated in the Declaration of Helsinki. Also, the participants and their attendants were briefed about the aim and objective of the study. Also, we obtained written consent from all the patients or their primary caregivers before participation.

## Result

### Demographics of study participants

The descriptive statistics for all the COVID-19 patients are summarized and presented in Table 1. Based on the severity of COVID-19 patients, we classified 198 patients as non-severe, and 108 patients were in a severe category. Of the 306 participants, 187 (61.11%) were male, and 119 (38.89%) were female. The age and BMI of our cohort were 48.79±8.53 years and 25.82±3.75 kg/m$^2$, respectively. After treatment, 294 (96.08%) patients were tested COVID negative in the RT-PCR test, and 12 (3.92%) patients had died. Among the participants, 73.40% of patients were above 40-years of age, and 36.60% of patients were below 40 years of age. Most of the participants were between 21–60 years old. Among the COVID-19 patients, two-thirds had normal BMI, and 77.45% had any comorbidity. Common comorbidities were diabetes (54.58%), hypertension (53.27%), and bronchial asthma (22%). Besides, we also observed that 19% had a history of any other comorbid diseases.

### Hematological abnormalities in COVID-19 patients

We presented the hematological alterations of the cohort in Table 2. First, we compared hematological variations between SCP and NSCP. The SCP had lower RBC and Hb levels than NSCP. The NSCP showed significantly lower WBC, neutrophils, platelet, CRP, ferritin, d-dimer, creatinine, SGPT, SGOT, and bilirubin levels than SCP. However, we observed the electrolyte levels (Na+, K+, Cl-, and HCO3) were higher in NSCP than SCP. We demonstrated the changes of hematological parameters in two different disease states using box-plot graphs (Fig 1).

### Relationships among hematological parameters

Pearson correlation coefficient were used for statistical analysis for correlating inter parameter variables. Relationships among the parameters of the cohort at different severity levels of COVID-19 are presented Table 3. Briefly, in SCP, Hb levels were positively associated with RBC (r = 0.456; p<0.001) and PCV (r = 0.593; p<0.001). RBC and PCV levels were positively correlated with each other in SCP group (r = 0.593; p<0.001). Whereas, PCV and d-dimer levels were negatively correlated with each other (r = -0.302; p<0.001). WBC and neutrophil levels were positively correlated with each other (r = 0.333; p<0.001), whereas, WBC levels were negatively correlated with lymphocyte (r = 0.400; p<0.001) and monocyte levels (r = 0.330;

**Table 1. Characteristics and comorbid diseases of COVID-19 patients.**

| Parameters | SCP (n = 108) | | | NSCP (n = 198) | | | p-value |
|---|---|---|---|---|---|---|---|
| | n | % | Mean ± SD | n | % | Mean ± SD | |
| Age in years | | | 54.07±9.82 | | | 45.55±6.34 | <0.01 |
| 20–40 | 24 | 22 | | 88 | 44 | | |
| 41–60 | 38 | 35 | | 66 | 33 | | |
| 61–80 | 46 | 43 | | 44 | 22 | | |
| Sex | | | | | | | <0.05 |
| Male | 92 | 85 | | 95 | 48 | | |
| Female | 16 | 15 | | 103 | 52 | | |
| BMI (kg/m$^2$) | | | 27.73±4.82 | | | 24.45±3.82 | <0.01 |
| Below 18.5 (CED) | 1 | 1 | | 7 | 4 | | |
| 18.5–25 (normal) | 63 | 58 | | 139 | 70 | | |
| Above 25 (obese) | 44 | 41 | | 52 | 26 | | |
| Presence of comorbid disease | | | | | | | <0.01 |
| Yes | 102 | 94 | | 135 | 68 | | |
| No | 6 | 6 | | 63 | 32 | | |
| Hypertension | | | | | | | <0.01 |
| Yes | 73 | 68 | | 90 | 45 | | |
| No | 35 | 32 | | 108 | 55 | | |
| Diabetes | | | | | | | |
| Yes | 79 | 73 | | 88 | 44 | | |
| No | 29 | 27 | | 110 | 56 | | |
| Bronchial Asthma | | | | | | | <0.05 |
| Yes | 38 | 35 | | 30 | 15 | | |
| No | 70 | 65 | | 168 | 85 | | |
| CKD | | | | | | | <0.01 |
| Yes | 3 | 3 | | 1 | 1 | | |
| No | 105 | 97 | | 197 | 99 | | |
| Outcome | | | | | | | <0.01 |
| COVID-19 survivors | 96 | 89 | | 198 | 100 | | |
| Death | 12 | 11 | | 0 | 0 | | |

SCP: Severe COVID-19 patient; NSCP: Non-severe COVID-19 patient; BMI: Body mass index; CED: Chronic energy deficiency; CKD: Chronic kidney disease.

p<0.001). Moreover, neutrophil levels were negatively correlated with Lymphocyte (r = -0.530; p<0.001) and monocyte (r = -0.354; p<0.001) levels. And CRP and creatinine were positively connected with each other (r = 0.330; p<0.001).

## Regression analysis

We performed binary logistic regression analysis to measure risk estimation of COVID-19 severity for demographic profile and comorbid diseases (Table 4). The COVID patients aged below 40 years were 1.060 times less likely to develop severe symptoms (OR = 1.060, 95% CI 1.025–1.095, <0.05) than patients aged 40 years and above. Male patients were 3.252 times more prone develop severe COVID-19 symptoms (OR = 3.252, 95% CI 1.001–15.140, <0.05). Obese subjects showed to be 1.130 times more susceptible to get severe COVID-19 (OR = 1.130, 95% CI 1.055–1.212, p<0.001) in comparison to healthy subjects. COVID-19 patients with any comorbid diseases were at 2.881 times higher risk to turn into severe COVID-19 than patients without comorbidity (OR = 2.881, 95% CI 1.364–22.727, <0.05).

**Table 2. Hematological parameters of the study cohort at different severity levels of COVID-19.**

| Parameter | Reference value | SCP (n = 108) Mean±SD | NSCP (n = 198) Mean±SD | P-value |
|---|---|---|---|---|
| Hb (g/dL) | Male: 13.8–17.2; Female: 12.1–15.1 | 12.61±5.98 | 12.68 ±1.36 | 0.012 |
| RBC (× $10^{12}$/L) | Male: 4.7–6.1; Female: 4.2–5.4 | 4.48±0.64 | 4.54±0.44 | <0.001 |
| PCV (%) | Male: 38.3–48.6; Female: 35.5–44.9 | 36.58±5.20 | 40.99±2.77 | <0.001 |
| MCV (fL) | 80–100 | 81.71±6.48 | 87.90±3.88 | <0.001 |
| WBC (× $10^9$/L) | Male: 3.9–10.6; Female: 3.5–11 | 8.30±3.65 | 7.32±1.98 | <0.001 |
| Neutrophils (%) | 40–60 | 72.54±11.27 | 64.51±9.19 | <0.001 |
| Lymphocytes (%) | 20–40 | 21.61±11.89 | 30.36±5.92 | <0.001 |
| Monocytes (%) | 2–8 | 4.13±1.95 | 4.48±1.36 | 0.005 |
| Eosinophils (%) | 1–4 | 1.27±1.58 | 2.96±1.51 | <0.001 |
| Basophils (%) | <1 | 0.38±0.36 | 0.42±0.38 | 0.049 |
| Platelet (× $10^9$/L) | 150–400 | 240.41±95.45 | 235.02±66.29 | 0.019 |
| CRP (mg/dL) | <10 | 6.55±7.25 | 0.19±0.11 | <0.001 |
| Ferritin (ng/mL) | Male: 20–250; Female: 12–263 | 651.86±793.30 | 86.41±34.78 | <0.001 |
| D-Dimer (ng/mL) | <250 | 756.73±600.08 | 239.00±99.25 | <0.001 |
| Creatinine (mg/dL) | 0.6–1.3 | 1.15±0.93 | 0.83±0.27 | <0.001 |
| Albumin (g/dL) | 3.5–5.5 | 3.32±0.55 | 4.16±0.51 | 0.259 |
| SGPT (U/L) | 7 to 56 | 58.04±45.12 | 36.86±12.67 | <0.001 |
| SGOT (U/L) | 5 to 40 | 39.34±29.32 | 28.15±8.14 | <0.001 |
| Bilirubin (mg/dL) | 0.3–1.0 | 0.49±0.27 | 0.46±0.27 | 0.049 |
| Na$^+$ (mmol/L) | 135–145 | 136.41±5.28 | 139.89±2.90 | <0.001 |
| K$^+$ (mmol/L) | 3.5–5.5 | 3.81±0.66 | 4.04±0.44 | <0.001 |
| Cl$^-$ (mmol/L) | 98–106 | 101.19±5.22 | 102.66±3.36 | <0.001 |
| HCO3 (mmol/L) | 23–29 | 26.58±3.89 | 28.05±2.29 | <0.001 |

$p<0.05$ (Significant difference between patient and control groups at 95% confidence interval). SCP, severe COVID-19 patients; NSCP, non-severe COVID-19 patients; Hb, hemoglobin; RBC, red blood cell; PCV, packed cell volume; MCV, mean corpuscular volume; WBC, white blood cell; CRP, carbon reactive protein, SGPT, serum glutamic pyruvic transaminase; SGOT, serum glutamic oxaloacetic transaminase; Na, sodium; K, potassium; CI, chloride; HCO3, bicarbonate.

COVID-19 patients with diabetes, hypertension, bronchial asthma, and CKD were 3.776, 2.409, 2.835, and 2.069 times more sensitive to develop severity than patients without these comorbid diseases (OR = 3.776, 95% CI 1.832–16.264, <0.05; OR = 2.409, 95% CI 1.693–9.090, <0.05; OR = 2.835, 95% CI 1.023–12.978, <0.05; OR = 2.069, 95% CI 1.011–10.422, p<0.001, respectively).

## Diagnostic performance evaluation of target parameters

For the analysis of significant differences of hematological parameters between SCP and NSCP, we used the receiver operating characteristic (ROC) curve analysis. Determination of the diagnostic performance is based on the area under the curve (AUC) as follows: AUC = 0.9–1.0, excellent; AUC = 0.8–0.9, good; AUC = 0.7–0.8, fair; AUC = 0.6–0.7, poor; and AUC <0.6, not useful [29, 30]. Among the parameters, CRP, d-dimer, and ferritin showed good diagnostic performances according to ROC analysis (Fig 2 and Table 5). CRP showed AUC value of 0.990 (95% CI 0.985–0.995) at a cutoff point 0.30 with 99.0% sensitivity and 79.6% specificity. Also, d-dimer showed an AUC value of 0.828 (95% CI 0.795–0.862) at a cut-off point 275.1 with 74.5% sensitivity and 64.9% specificity. Moreover, ferritin showed AUC value of 0.997 (95% CI 0.994–0.999; p<0.001) at a cutoff point 120.5 with 99.7% sensitivity and 82.0% specificity.

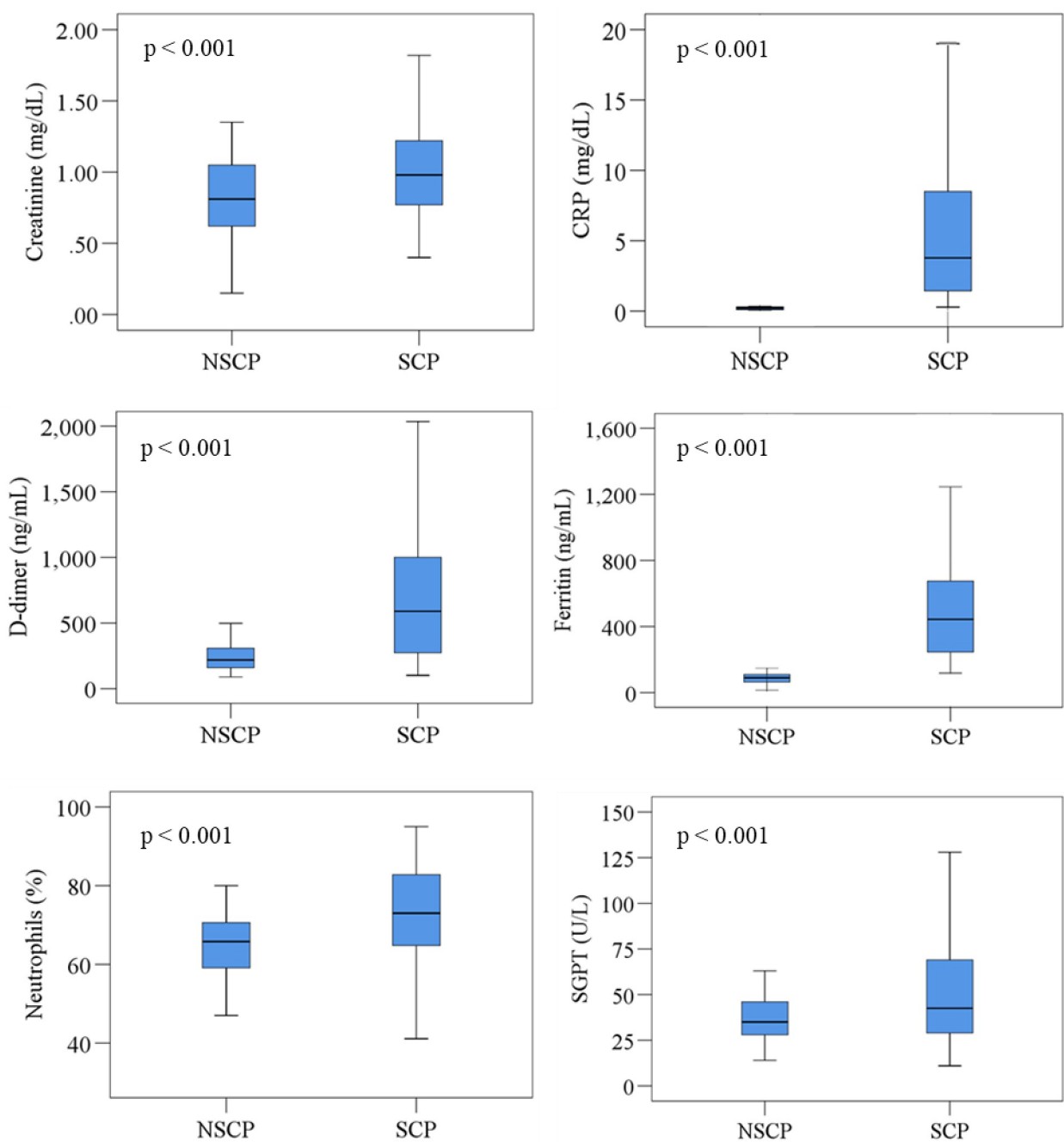

**Fig 1. Changes of blood parameters in two different disease states.** Independent sample t-test was applied and p < 0.05 was considered as significant at 95% confidence of interval. NSCP, non-severe COVID-19 patients; SCP, severe COVID-19 patients.

## Discussion

We conducted this observational cohort study to assess the connection of hematological abnormalities with the severity of hospitalized COVID-19 patients with complications. Amidst the increasing rate of COVID-19 transmission [31], it is vital to generate comprehensive information regarding the COVID-19 severity to measure the mortality risks. Hematological

**Table 3. Correlations among research parameters in the cohort at different severity stages of COVID-19.**

| Parameter | SCP (n = 108) | | NSCP (n = 198) | |
|---|---|---|---|---|
| | r | p | r | p |
| Hb and RBC | 0.456 | <0.001 | 0.053 | 0.353 |
| Hb and PCV | 0.593 | <0.001 | -0.040 | 0.486 |
| RBC and PCV | 0.506 | <0.001 | 0.025 | 0.665 |
| PCV and D-dimer | -0.302 | <0.001 | 0.051 | 0.374 |
| WBC and Neutrophils | 0.333 | <0.001 | 0.028 | 0.627 |
| WBC and Lymphocytes | -0.400 | <0.001 | 0.074 | 0.196 |
| WBC and Monocytes | -0.330 | <0.001 | -0.016 | 0.778 |
| Neutrophils and Lymphocytes | -0.530 | <0.001 | -0.050 | 0.381 |
| Neutrophils and Monocytes | -0.354 | <0.001 | -0.030 | 0.601 |
| CRP and Creatinine | 0.330 | <0.001 | -0.031 | 0.588 |
| SGPT and SGOT | 0.479 | <0.001 | -0.112 | 0.051 |
| $Na^+$ and $Cl^-$ | 0.455 | <0.001 | -0.008 | 0.889 |

r = Correlation co-efficient; p = Significance; Negative values specify opposite correlation. Correlation is significant at 0.05 level (two-tailed). SCP, severe COVID-19 patients; NSCP, non-severe COVID-19 patients; Hb, hemoglobin; RBC, red blood cell; PCV, packed cell volume; WBC, white blood cell; CRP, carbon reactive protein; SGPT, serum glutamic pyruvic transaminase; SGOT, serum glutamic oxaloacetic transaminase; Na, sodium; Cl, chloride.

parameters and comorbid disease can help to measure COVID-19 severity [32]. Therefore, we can assess patient's severity and mortality risks by easily monitoring those potential indicators. In our findings, we analyzed some hematological parameters in different severity stages of hospitalized COVID-19 patients. Among the parameters, we found a decreased level of Hb, RBC, packed cell volume (PCV), mean corpuscular volume (MCV), lymphocytes, eosinophils, and increased level of WBC, neutrophils in SCP than NSCP. Also, we observed increased CRP, ferritin, d-dimers, SGOT, and SGPT levels and decreased $Na^+$, $K^+$, $Cl^-$, and $HCO_3$ levels in SCP than NSCP. However, the values of most parameters were within normal ranges. Therefore, these parameters might have statistical significance but not clinical significance. According to our knowledge, this is the first-ever study in Bangladesh to find an association of hematological abnormalities in COVID-19 patients with diagnostic performance analysis. Also, we validated the present study findings using ROC analysis which was absent in most earlier studies. Inconsistent with our findings, speculated evidence indicates that hematological abnormalities are very prevalent in SCP and increases the requirement of hospital stay and ICU support

**Table 4. Univariate and multivariate analysis of risk factors associated with COVID-19 severity.**

| Characteristics of patients | Univariate analysis of risk factors | | | Multivariate analysis of risk factors | | |
|---|---|---|---|---|---|---|
| | SCP (n = 108) | NSCP (n = 198) | p-value | OR | 95% CI | p-value |
| Age above 40 years | 84 (78) | 110 (56) | <0.01 | 1.060 | 1.025–1.095 | <0.05 |
| Male sex | 92 (85) | 95 (48) | <0.05 | 3.258 | 1.001–15.140 | <0.05 |
| Patient with obesity | 44 (41) | 52 (26) | <0.01 | 1.130 | 1.055–1.212 | <0.001 |
| Patient with any comorbid diseases | 102 (94) | 135 (68) | <0.01 | 2.881 | 1.364–22.727 | <0.05 |
| Patients with diabetes | 79 (73) | 88 (44) | <0.01 | 3.776 | 1.832–16.264 | <0.05 |
| Patients with hypertension | 73 (68) | 90 (45) | <0.01 | 2.409 | 1.693–9.090 | <0.05 |
| Patients with bronchial asthma | 38 (35) | 30 (15) | <0.05 | 2.835 | 1.023–12.978 | <0.05 |
| Patients with CKD | 3 (3) | 1 (1) | <0.01 | 2.069 | 1.011–10.422 | <0.001 |

SCP: Severe COVID-19 patient; NSCP: Non-severe COVID-19 patient; CKD: Chronic kidney disease.

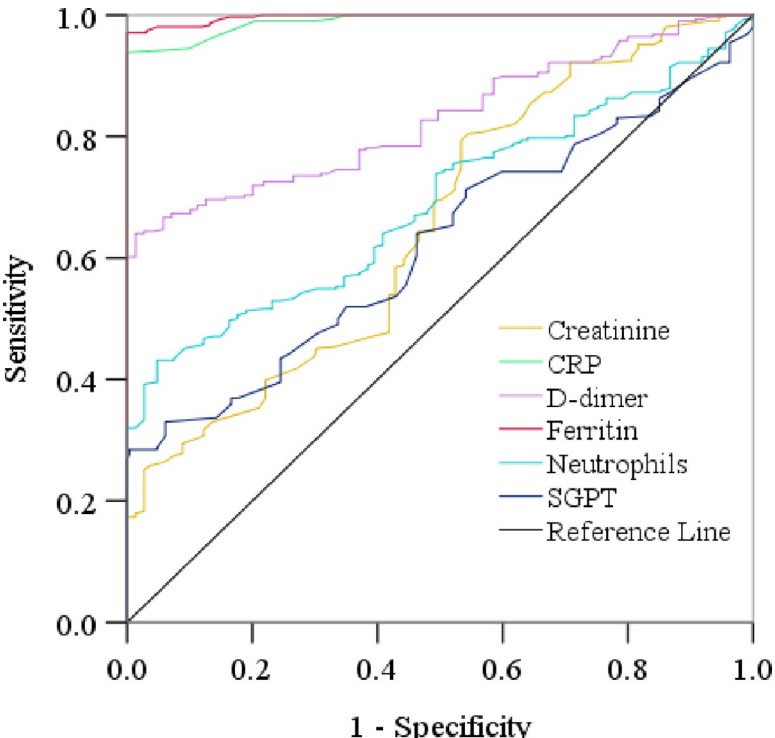

**Fig 2. Receiver operating characteristic (ROC) curve is showing the relative diagnostic performances of creatinine, CRP, D-dimer, ferritin, neutrophils, and SGPTA.** The cut-off points were detected for creatinine, CRP, d-dimer, ferritin, neutrophils, and SGPT as 0.30 mg/dL, 275.10 mg/dL, 120.5 ng/mL, 68.0ng/mL, and 40.5U/L, respectively.

otherwise accelerate lethality. Different hematologic parameter level irregularities are associated with the risk of COVID-19 severity [33–35]. Terpos et al. reviewed hematological parameters in COVID patients and observed some abnormalities in SCP than NSCP [36]. Also, some other studies supported the present findings describing the evidence of the correlation between elevated CRP and ferritin levels in SCP [37–39].

After the SARS-CoV-2 entrance into the blood, it primarily affects the angiotensin-converting enzyme (ACE2), a receptor of the SARS-CoV-2 expressed in various organs like the liver, heart, gastrointestinal tract. After 7–14 days of infection, the CT scan report shows changes in

**Table 5. Receiver operating characteristic analysis of promising markers for severity of COVID-19.**

| Parameters | Area under the curve | Asymptotic Significance | Asymptotic 95% Confidence Interval | | Cutoff value | Sensitivity | Specificity |
|---|---|---|---|---|---|---|---|
| | | | Lower Bound | Upper Bound | | | |
| Creatinine | 0.653 | <0.001 | 0.610 | 0.697 | 0.85 | 64.4% | 54.4% |
| CRP | 0.990 | <0.001 | 0.984 | 0.995 | 0.30 | 99.0% | 79.6% |
| D-dimer | 0.828 | <0.001 | 0.795 | 0.862 | 275.1 | 74.5% | 64.9% |
| Ferritin | 0.997 | <0.001 | 0.994 | 0.999 | 120.5 | 99.7% | 82.0% |
| Neutrophils | 0.688 | <0.001 | 0.645 | 0.731 | 68.0 | 62.1% | 59.2% |
| SGPT | 0.620 | <0.001 | 0.575 | 0.665 | 40.5 | 52.0% | 61.9% |

$p < 0.05$ (Significant difference between the groups at 95% confidence interval).

CRP, carbon reactive protein, SGPT, serum glutamic pyruvic transaminase.

the worsening situation. At this stage, lymphocyte count decreases, and inflammatory cytokine increases that deteriorate the condition of patients [40]. Lymphopenia is common in critically ill patients that correlate with the severity of COVID-19 [35, 37, 41]. Moreover, along with the increased leukocytes, reduced basophils, monocytes, lymphocytes, eosinophils, and platelets are frequently observed in SCP [42, 43]. According to studies, neutrophilia occurs in COVID-19 patients who required ICU support. Therefore, the neutrophil-to-lymphocyte ratio can serve as an indicator for the severity of COVID-19 patients [34, 44]. Several studies observed the incidence of thrombocytopenia due to decreased platelet count in a significant portion of patients who needed to admit to the hospital [35, 37]. Lippi et al. accumulated four different studies where it has been showing that the Hb level of COVID patients is in descending tendency and a contributor for worse advancement [45]. A meta-analysis of 21 studies on 3,377 patients reported the significant association of abnormal hematological parameters among SCP [46]. So, the hematological abnormalities have a close association with the severity of COVID-19, prolongation of hospitalization, and the requirement of ICU supports.

Based on the above discussion, we noticed that several previous studies recognized the hematological abnormalities in SCP. But we cannot use these changes for the assessment of COVID severity due to their absence of predictive performance [34–39]. Initial alterations of any blood parameters may not serve as a diagnostic predictor until these changes not validated for sensitivity and specificity. In the present study, we observed a lot of parameters altered in blood levels. But only CRP and ferritin showed good predicting performances for COVID-19 severity based on the diagnostic performance evaluation by ROC analysis. These three parameters demonstrated good sensitivity and specificity. Therefore, we recommend these three factors together can serve as differentiation factors for SCP and NSCP.

Moreover, coagulation frequently occurs among SCP [38, 47]. Coagulation is associated with a higher level of d-dimer and prothrombin time prolongation. These are frequently encountered in severe stages of COVID-19 and are also considered a prime reason for death [34, 47]. Many past studies have presumed the elevated d-dimer levels responsible for altered coagulation in COVID-19 patients [35, 48–50]. Thus d-dimer, a fibrin degradation product, is a possible predictive factor for COVID-19 severity. In the early stage of the COVID-19 pandemic, a study found elevated d-dimer levels in 260 COVID-19 patients out of 560 in China [35]. Also, scientific evidence revealed that patients who died in COVID-19 had a higher level of d-dimer [51]. Besides, significant increased d-dimer levels were in ICU patients or patients with critical conditions in different studies [34, 42]. However, the increased level of d-dimer found in hospitalized COVID patients had a significant predictive performance for the COVID severity in our country. Changes in the immune system and lung damage are some of the prime abnormalities in COVID-19 patients. It is well established that cytokine storm causes to increase several inflammatory biomarker's levels in SCP [52, 53]. Another cohort study showed that the augmented risks of developing ARDS in SCP were associated with elevated ferritin, CRP, and other biomarkers levels [54]. Ferritin is a protein that stores iron. The increased serum ferritin levels reflect the iron level. Another study revealed that patients who died in COVID-19 had higher ferritin levels and had to stay in hospital for a longer time [55]. Therefore, elevated serum ferritin induces cytokine storms and severity of COVID-19 patients [56, 57]. Thus, ferritin levels can serve as a factor for monitoring COVID-19 severity [46]. A study reported high levels of CRP in SCP. The higher levels of CRP may cause cytokine storms and affect liver function. Therefore, hepatic abnormalities worse the situation of SCP [58]. Ferritin is an independent risk factor for the severity of COVID-19. However, there is a positive relationship between ferritin and CRP in COVID fatality [59]. CRP is a marker of severe infections and inflammatory responses [60, 61]. Many past studies reported the association between CRP levels and lung lesions in the severe stage of hospitalized COVID-19 patients [62, 63].

Additionally, COVID-19 is a systemic infection responsible for several clinical appearances [64]. In the present study, we observed that age, sex, obesity, and comorbid diseases were significantly associated with the risks for developing severity among COVID-19 patients. Several retrospective studies also demonstrated increased mortality and morbidity rates in older adults with underlying comorbidities. Myocardia injuries are also a consequence of hematological abnormalities. The recent data supports abnormal levels of leukocyte, lymphocyte, platelet counts were predominant in patients with COVID-19 who had myocardia injuries [65]. One study found that associated comorbidities and recent history of chemotherapy can potentiate viral load. Therefore, the death rate was 35% for those COVID-19 patients [66]. Another study conducted in the USA showed that most COVID-19 patients with comorbid diseases were treated in the ICU by mechanical ventilation [67]. A French study found that COVID-19 patients with abnormal BMI had 7-fold higher chances to get admitted into the hospitals for ICU support than non-obese patients [68].

Moreover, several studies showed no significant associations of CRP with the incidence of diabetes [69, 70]. Other studies reported a high-CRP association only with diabetes-induced complications, like nephropathy and cardiovascular risk. Ethnic group differences were evident in detecting the association of CRP levels with hypertension [71]. No correlation was found between CRP and hypertension levels in Bangladeshi [72], and Chinese participants, except Hispanic participants [73]. A study showed that an increased CRP level would not result in a higher CKD risk [74]. Evidence reported an association of CRP levels with the severity of asthma and COPD. We found a general increase of CRP in all SCP with or without comorbid diseases compared to NSCP, suggesting that CRP is a crucial factor in determining the severity of COVID patients [75, 76]. Patients with diabetes had generally higher d-dimer levels. A study showed increased plasma d-dimer levels in patients with impaired fasting glucose [77], which is because of developing a hypercoagulable state [78]. Increased level of d-dimer was reported in patients with hypertension [79], CKD [80], and COPD [81]. Increased level of ferritin predicts the severity of COVID-19 diseases. A high level of ferritin is independently associated with the prevalence of diabetes [82], hypertension [83], CKD [84], COPD, and worse asthma symptoms because of a strong correlation with systemic inflammation. The present found an increase of ferritin levels in SCP regardless of comorbid diseases. The increased ferritin levels may worsen COVID-19 symptoms by contributing to the cytokine storm.

## Potential limitations of the study

The present study has few drawbacks. The study included COVID-19 patients from Dhaka city only. Also, the subsequent follow-up information was not available with this study. We collected the clinical outcome data from only 52.7% of admitted COVID-19 patients from the respective hospital during study time. The exact mechanism of hematological alterations in COVID-19 patients is not well-explained yet. Therefore, we recommend further studies regarding the hematological abnormalities in COVID-19 to identify the actual causes.

## Conclusion

Physicians continuously monitor several blood parameters to measure the severity and mortality risks of COVID-19 patients. Among the altered hematological parameters, elevated CRP and ferritin levels might use as predictable markers to assess the COVID-19 severity and mortality risks. Also, these parameters might help to evaluate the treatment plan and decision at the hospital setup. Therefore, we recommend these parameters as factors for early detection of COVID-19 severity that facilitates the treatment decisions. This topic needs to be explored

further for quick initiation of treatments for SCP and optimization of COVID-19 treatment at hospital care.

## Acknowledgments

We thank all the participants and their primary care givers for their cooperation to this study. Also, we thank the physicians and administrative staffs of the COVID unit of Evercare Hospital Ltd, Dhaka, Bangladesh, for their support to this study.

## Author Contributions

**Conceptualization:** Md. Ashrafur Rahman, Yeasna Shanjana, Md. Ismail Tushar, Tarif Mahmud, Md. Rabiul Islam, Hasan Mahmud Reza.

**Data curation:** Md. Ashrafur Rahman, Ghazi Muhammad Sayedur Rahman, Zahid Hossain Milan, Tamanna Sultana, Ali Mohammed Lutful Hoq Chowdhury.

**Formal analysis:** Md. Ashrafur Rahman, Mohiuddin Ahmed Bhuiyan, Md. Rabiul Islam.

**Investigation:** Yeasna Shanjana, Zahid Hossain Milan, Tamanna Sultana, Ali Mohammed Lutful Hoq Chowdhury.

**Methodology:** Mohiuddin Ahmed Bhuiyan, Md. Rabiul Islam.

**Project administration:** Ghazi Muhammad Sayedur Rahman, Hasan Mahmud Reza.

**Supervision:** Md. Rabiul Islam, Hasan Mahmud Reza.

**Writing – original draft:** Md. Ashrafur Rahman, Md. Ismail Tushar, Tarif Mahmud.

**Writing – review & editing:** Md. Ashrafur Rahman, Md. Rabiul Islam, Hasan Mahmud Reza.

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
