## [Decision Letter · Decision Letter 0]

29 Jun 2021

PONE-D-21-19091

Hematological abnormalities and comorbidities are associated COVID-19 related deaths among hospitalized patients: Experience from Bangladesh

PLOS ONE

Dear Dr. Islam,

Thank you for submitting your manuscript to PLOS ONE. After careful consideration, we feel that it has merit but does not fully meet PLOS ONE’s publication criteria as it currently stands. Therefore, we invite you to submit a revised version of the manuscript that addresses the points raised during the review process.

Please address the issues and revise accordingly.

We look forward to receiving your revised manuscript.

Kind regards,

Academic Editor

PLOS ONE

Journal Requirements:

[Also, we thank the physicians and administrative staffs of the COVID unit of Evercare Hospital Ltd, Dhaka, Bangladesh, for their support to this study.]

 [The author(s) received no specific funding for this work.]

Additionally, because some of your funding information pertains to [commercial funding//patents], we ask you to provide an updated Competing Interests statement, declaring all sources of commercial funding.

In your Competing Interests statement, please confirm that your commercial funding does not alter your adherence to PLOS ONE Editorial policies and criteria by including the following statement: "This does not alter our adherence to PLOS ONE policies on sharing data and materials.” as detailed online in our guide for authors  http://journals.plos.org/plosone/s/competing-interests.  If this statement is not true and your adherence to PLOS policies on sharing data and materials is altered, please explain how.

Please include the updated Competing Interests Statement and Funding Statement in your cover letter. We will change the online submission form on your behalf.

Reviewers' comments:

Reviewer's Responses to Questions

**Comments to the Author**

1. Is the manuscript technically sound, and do the data support the conclusions?

Reviewer #1: No

Reviewer #2: Partly

2. Has the statistical analysis been performed appropriately and rigorously? 

Reviewer #1: No

Reviewer #2: Yes

3. Have the authors made all data underlying the findings in their manuscript fully available?

Reviewer #1: Yes

Reviewer #2: Yes

4. Is the manuscript presented in an intelligible fashion and written in standard English?

Reviewer #1: Yes

Reviewer #2: Yes

5. Review Comments to the Author

Reviewer #1: 1. I think patient stratification is the primary concern in the current study. The SCP and NSCP are the outcomes. When the authors stratified patients according to their outcomes, their lab parameters must be different.

2. Table 2 shows the comparisons of hematological parameters between the SCP and NSCP groups. We can see many items with statistical significance (ex. Hb, RBC, WBC, Platelet, liver functions…..). However, the values of these parameters were all within normal ranges, suggesting these parameters might have statistical significance but not clinical significance.

3. In table 3, the authors analyzed the correlation among laboratory parameters and different disease severity. Even the p values were < 0.05 in some analyses, the correlation coefficient value was low, indicating the limited clinical applications.

4. On line 200, page 13, the authors analyzed the risk factors associated with COVID-19 “outcome.” I would suggest not using the word “outcome” but “severity.”

5. In Table 4, I would suggest the authors identify the risk factors by univariate analysis first and put substantial elements found by univariate analysis into multivariate analysis.

6. In Table 5, the sensitivity and specificity of creatinine, D-dimer, neutrophil counts, and GPT were low, suggesting these parameters are NOT promising markers for COVID-19 severity.

Reviewer #2: Major comments:

1. I suggest it is better to compare the parameters between SCP and NSCP in Table 1.

2. Since you compare SCP and NSCP but not survivors and COVID-19-derived death (3.92% in this study), title “associated COVID-19 related deaths” is not appropriate.

3. Please show normal range of the parameter in Table 2. For example, WBCs ranged 3.9~10.6 for male and 3.5~11 for female are normal.

4. Please only show significant correlation r>0.3 and <-0.3 in Table3 (r=-0.3~0.3: poor correlation; r: 0.3~0.6 and -0.3~-0.6: media correlation; r=0.6~0.9 and -0.6~-0.9: high correlation; r=1 and -1: complete correlation. The comparison between individual parameter can be shown as supplemental data.

5. Please show statistical value in Figure 1.

6. Did increase of CRP, d-dimer, ferritin associate with any comorbidity disease.

6. PLOS authors have the option to publish the peer review history of their article (what does this mean?). If published, this will include your full peer review and any attached files.

Reviewer #1: No

Reviewer #2: No

---

## [Author Response · Author response to Decision Letter 0]

7 Jul 2021

Dear Editors and Reviewers,

Thank you for your letter and the reviewers' comments on our manuscript entitled "Hematological abnormalities and comorbidities are associated COVID-19 related deaths among hospitalized patients: Experience from Bangladesh" (Manuscript ID PONE-D-21-19091). All the comments were valuable and helpful to the revision and improvement of the manuscript. We have carefully studied the comments and made corrections, which we hope will merit your approval. We marked the revised portions using track changes. Our point-by-point answers to the reviewers’ comments appear at the end of this letter.

We earnestly appreciate the Editors'/Reviewers' work. We hope that after this revision, the paper will be deemed fit for publication. We would be glad to respond to any further questions and comments that you may have. 

Once again, thank you very much for your comments and suggestions.

Best regards,

Md. Rabiul Islam, PhD

Assistant Professor, Department of Pharmacy, University of Asia Pacific, 74/A Green Road, Farmgate, Dhaka-1215, Bangladesh. Email: robi.ayaan@gmail.com; Cell: +8801916031831

Point by point authors’ responses to the reviewers

Manuscript ID PONE-D-21-19091

Title: Hematological abnormalities and comorbidities are associated COVID-19 related deaths among hospitalized patients: Experience from Bangladesh

Reviewer #1

Comment 1: I think patient stratification is the primary concern in the current study. The SCP and NSCP are the outcomes. When the authors stratified patients according to their outcomes, their lab parameters must be different.

Author responses

Thank you for your review and valuable comments on this manuscript.

In the present study, the severity of covid-19 patients was assessed by 1. patients admitted in hospital with confirmed pneumonia by lung CT, 2. patients with respiratory distress (rate > 30 breaths/min) and oxygen capacity level < 93%, and 3. patients required intensive care unit (ICU) support or mechanical ventilation. It would be better if we could perform patient stratification asses their severity. We are sorry for not systematically performing the risk stratification of covid patients for categorizing them into SCP and NSCP based on their health status and other factors. Therefore, we mentioned this as a limitation of the present study in the revised manuscript.

Comment 2: Table 2 shows the comparisons of hematological parameters between the SCP and NSCP groups. We can see many items with statistical significance (ex. Hb, RBC, WBC, Platelet, liver functions….). However, the values of these parameters were all within normal ranges, suggesting these parameters might have statistical significance but not clinical significance.

Author responses

Thank you for your valuable observation. All the parameters were within normal ranges in NSCP but ferritin and d-dimer were out of range in SCP. Following your comment, we have mentioned this observation in our revised (in highlighted manuscript: page 17, line 259-261).

Comment 3: In table 3, the authors analyzed the correlation among laboratory parameters and different disease severity. Even the p values were < 0.05 in some analyses, the correlation coefficient value was low, indicating the limited clinical applications.

Author responses

Thank you for your valuable observation. Following your observation, we have now removed all poor correlations (r<0.3 and r>-0.3) from Table 3 in the revised manuscript even the p values were < 0.05. Now revised Table 3 is showing significant correlation among the parameters (in highlighted manuscript: page 12). 

Comment 4: On line 200, page 13, the authors analyzed the risk factors associated with COVID-19 “outcome.” I would suggest not using the word “outcome” but “severity.”

Author responses

Thank you for your suggestion. We have revised this line. Also, we have replaced “outcome” in other places with similar meaning (in highlighted manuscript: page 14, line 210, Title in Table 4, page 14).

Comment 5: In Table 4, I would suggest the authors identify the risk factors by univariate analysis first and put substantial elements found by univariate analysis into multivariate analysis.

Author responses

Thank you so much for your nice suggestions. We have now performed univariate analysis and shown risk factors from both univariate and multivariate analyses in Table 4 in the revised manuscript (in highlighted manuscript: Table 4, page 14).

Comment 6: In Table 5, the sensitivity and specificity of creatinine, D-dimer, neutrophil counts, and GPT were low, suggesting these parameters are NOT promising markers for COVID-19 severity.

Author responses

Thank you so much for your valuable suggestions in this regard. Based on our findings, previously we suggested elevated CRP, ferritin, and d-dimer as promising markers for COVID-19 severity. Following your observation, we have made necessary corrections according to sensitivity and specificity values. In the revised manuscript, we suggested elevated CRP and ferritin levels as promising markers for COVID-19 severity (in highlighted manuscript: page 19, lines 293; page 22, line 363). 

Reviewer #2

Comment 1: I suggest it is better to compare the parameters between SCP and NSCP in Table 1.

Author responses

Thank you so much for your valuable suggestions in the point. We have now revised Table 1 comparing parameters between SCP and NSCP (in highlighted manuscript: page 8).

Comment 2: Since you compare SCP and NSCP but not survivors and COVID-19-derived death (3.92% in this study), title “associated COVID-19 related deaths” is not appropriate.

Author responses

Thank you so much for your excellent points regarding the title of the present study. Following your observation, we have now revised the title as below-

Hematological abnormalities and comorbidities are associated with COVID-19 severity among hospitalized patients: Experience from Bangladesh 

Comment 3: Please show normal range of the parameter in Table 2. For example, WBCs ranged 3.9~10.6 for male and 3.5~11 for female are normal.

Author responses

Thank you for your observation. We have now shown normal range of the parameter in Table 2 in revised Table 2 (in highlighted manuscript: page 10).

Comment 4: Please only show significant correlation r>0.3 and <-0.3 in Table3 (r=-0.3~0.3: poor correlation; r: 0.3~0.6 and -0.3~-0.6: media correlation; r=0.6~0.9 and -0.6~-0.9: high correlation; r=1 and -1: complete correlation. The comparison between individual parameter can be shown as supplemental data.

Author responses

Thank you for your valuable observation. Following your observation, we have now removed all poor correlations (r<0.3 and r>-0.3) from Table 3 in the revised manuscript even the p values were < 0.05. Now revised Table 3 is showing significant correlation among the parameters. We believe the comparison between individual parameter which are not significant would add any value for this article. Therefore, we discarded them from the revised Table 3 (in highlighted manuscript: page 12). Also, we made necessary corrections according to revised table 3 in the text (in highlighted manuscript: page 11, lines 188-197).

Comment 5: Please show statistical value in Figure 1.

Author responses

Thank you so much for this insightful suggestion. We have now added all the p-values in Figure 1.

Comment 6: Did increase of CRP, d-dimer, ferritin associate with any comorbidity disease?

Author responses

Thank you for your observation. We have now added information regarding the association of increased CRP, d-dimer, ferritin levels with comorbidity diseases that were found in COVID-19 patients. The revision is as follows- (in highlighted manuscript: page 21, lines 336-353)

Moreover, several studies showed no significant associations of CRP with the incidence of diabetes [69-70]. Other studies reported a high-CRP association only with diabetes-induced complications, like nephropathy and cardiovascular risk. Ethnic group differences were evident in detecting the association of CRP levels with hypertension [71]. No correlation was found between CRP and hypertension levels in Bangladeshi [72], and Chinese participants, except Hispanic participants [73]. A study showed that an increased CRP level would not result in a higher CKD risk [74]. Evidence reported an association of CRP levels with the severity of asthma and COPD. We found a general increase of CRP in all SCP with or without comorbid diseases compared to NSCP, suggesting that CRP is a crucial factor in determining the severity of COVID patients [75, 76]. Patients with diabetes had generally higher d-dimer levels. A study showed increased plasma d -dimer levels in patients with impaired fasting glucose [77], which is because of developing a hypercoagulable state [78]. Increased level of d-dimer was reported in patients with hypertension [79], CKD [80], and COPD [81]. Increased level of ferritin predicts the severity of COVID-19 diseases. A high level of ferritin is independently associated with the prevalence of diabetes [82], hypertension [83], CKD [84], COPD, and worse asthma symptoms because of a strong correlation with systemic inflammation. The present found an increase of ferritin levels in SCP regardless of comorbid diseases. The increased ferritin levels may worsen COVID-19 symptoms by contributing to the cytokine storm.

---

## [Decision Letter · Decision Letter 1]

15 Jul 2021

Hematological abnormalities and comorbidities are associated with COVID-19 severity among hospitalized patients: Experience from Bangladesh

PONE-D-21-19091R1

Dear Dr. Islam,

We’re pleased to inform you that your manuscript has been judged scientifically suitable for publication and will be formally accepted for publication once it meets all outstanding technical requirements.

Kind regards,

Academic Editor

PLOS ONE

Additional Editor Comments (optional):

Reviewers' comments:

Reviewer's Responses to Questions

**Comments to the Author**

1. If the authors have adequately addressed your comments raised in a previous round of review and you feel that this manuscript is now acceptable for publication, you may indicate that here to bypass the “Comments to the Author” section, enter your conflict of interest statement in the “Confidential to Editor” section, and submit your "Accept" recommendation.

Reviewer #1: All comments have been addressed

Reviewer #2: All comments have been addressed

2. Is the manuscript technically sound, and do the data support the conclusions?

Reviewer #1: Yes

Reviewer #2: Yes

3. Has the statistical analysis been performed appropriately and rigorously? 

Reviewer #1: Yes

Reviewer #2: Yes

4. Have the authors made all data underlying the findings in their manuscript fully available?

Reviewer #1: Yes

Reviewer #2: Yes

5. Is the manuscript presented in an intelligible fashion and written in standard English?

Reviewer #1: Yes

Reviewer #2: Yes

6. Review Comments to the Author

Reviewer #1: The authors have addressed all my comments adequately. The limitations of this study have been further mentioned as well. I think the paper is acceptable in the current version. Great work!

Reviewer #2: (No Response)

7. PLOS authors have the option to publish the peer review history of their article (what does this mean?). If published, this will include your full peer review and any attached files.

Reviewer #1: **Yes: **Chieh-Lin Jerry Teng

Reviewer #2: No

---

## [Editor Report · Acceptance letter]

19 Jul 2021

PONE-D-21-19091R1 

Hematological abnormalities and comorbidities are associated with COVID-19 severity among hospitalized patients: Experience from Bangladesh 

Dear Dr. Islam:

I'm pleased to inform you that your manuscript has been deemed suitable for publication in PLOS ONE. Congratulations! Your manuscript is now with our production department. 

Kind regards, 

on behalf of

Dr. Robert Jeenchen Chen 

Academic Editor

PLOS ONE